# Assessing the Intense Influenza A(H1N1)pdm09 Epidemic and Vaccine Effectiveness in the Post-COVID Season in the Russian Federation

**DOI:** 10.3390/v15081780

**Published:** 2023-08-21

**Authors:** Anna Sominina, Daria Danilenko, Andrey B. Komissarov, Maria Pisareva, Artem Fadeev, Nadezhda Konovalova, Mikhail Eropkin, Polina Petrova, Alyona Zheltukhina, Tamila Musaeva, Veronika Eder, Anna Ivanova, Kseniya Komissarova, Kirill Stolyarov, Ludmila Karpova, Elizaveta Smorodintseva, Anna Dorosh, Vera Krivitskaya, Elena Kuznetzova, Victoria Majorova, Ekaterina Petrova, Anastassia Boyarintseva, Andrey Ksenafontov, Anna Shtro, Julia Nikolaeva, Mikhail Bakaev, Elena Burtseva, Dmitry Lioznov

**Affiliations:** 1Smorodintsev Research Institute of Influenza, 197376 Saint Petersburg, Russia; daria.danilenko@influenza.spb.ru (D.D.); elena.kuznetsova@influenza.spb.ru (E.K.);; 2National Research Center for Epidemiology and Microbiology Named after N.F. Gamaleya, 123098 Moscow, Russia; 3Department of Infectious Diseases, First Pavlov State Medical University, 197022 Saint Petersburg, Russia

**Keywords:** influenza, SARS CoV-2, monitoring, incidence, hospitalization, molecular detection, antigenic properties, genetic analysis, susceptibility to antivirals

## Abstract

The COVID-19 pandemic had a profound impact on influenza activity worldwide. However, as the pandemic progressed, influenza activity resumed. Here, we describe the influenza epidemic of high intensity of the 2022–2023 season. The epidemic had an early start and peaked in week 51.2022. The extremely high intensity of the epidemic may have been due to a significant decrease in herd immunity. The results of PCR-testing of 220,067 clinical samples revealed that the influenza A(H1N1)pdm09 virus dominated, causing 56.4% of positive cases, while A(H3N2) influenza subtype accounted for only 0.6%, and influenza B of Victoria lineage—for 34.3%. The influenza vaccine was found to be highly effective, with an estimated effectiveness of 92.7% in preventing admission with laboratory-confirmed influenza severe acute respiratory illness (SARI) cases and 54.7% in preventing influenza-like illness/acute respiratory illness (ILI/ARI) cases due to antigenic matching of circulated viruses with influenza vaccine strains for the season. Full genome next-generation sequencing of 1723 influenza A(H1N1)pdm09 viruses showed that all of them fell within clade 6B.1A.5.a2; nine of them possessed H275Y substitution in the NA gene, a genetic marker of oseltamivir resistance. Influenza A(H3N2) viruses belonged to subclade 3C.2a1b.2a.2 with the genetic group 2b being dominant. All 433 influenza B viruses belonged to subclade V1A.3a.2 encoding HA1 substitutions A127T, P144L, and K203R, which could be further divided into two subgroups. None of the influenza A(H3N2) and B viruses sequenced had markers of resistance to NA inhibitors. Thus, despite the continuing circulation of Omicron descendant lineages, influenza activity has resumed in full force, raising concerns about the intensity of fore coming seasonal epidemics.

## 1. Introduction

Seasonal influenza viruses evolve continuously and cause severe disease annually, particularly in the elderly, children, pregnant women, and people with underlying chronic conditions. An average of 389,000 (uncertainty range 294,000–518,000) respiratory deaths were associated with influenza globally each year during 2002–2011, corresponding to ~2% of all annual respiratory deaths. Of these, 67% were among people 65 years and older [1]. Seasonal influenza vaccination remains the leading strategy for preventing infection and hospitalization with severe acute respiratory illness (SARI) caused by the influenza virus. While vaccination cannot completely prevent influenza epidemics, it does reduce the incidence of SARI requiring hospitalization, ultimately providing cost-effective protection and leading to a decrease in mortality in at-risk groups [2,3]. The effectiveness of current influenza vaccines largely depends on the degree of antigenic matching between vaccine strains and the circulating influenza viruses in a given season [4]. However, predicting the direction, mutation rate, and degree of drift of influenza viruses remains challenging, highlighting the need for continuous virological surveillance on a global scale [5]. Comprehensive surveillance of influenza in different countries is necessary to clarify unresolved issues, including the impact of vaccine strain mismatch on vaccine effectiveness and the relationship between anti-influenza population immunity and subsequent epidemic intensity.

The World Health Organization has implemented the Global Influenza Strategy (2019–2030) [6] to strengthen countries’ capacities and preparedness for future pandemics by enhancing seasonal prevention and control of influenza. Two key objectives of this strategy are to improve global influenza surveillance, monitoring, and data utilization, as well as to promote research and innovation. However, the COVID-19 pandemic significantly impacted the circulation of human seasonal influenza viruses worldwide and had a negative effect on influenza surveillance [7]. Nevertheless, the “End-to-end integration of SARS-CoV-2 and influenza sentinel surveillance: revised interim guidance” provided a solid foundation for harmonizing influenza and SARS-CoV-2 surveillance within the Global Influenza Surveillance and Response Network and its National Influenza Centers worldwide [8].

In the Russian Federation, there are two National Influenza Centers that conduct year-round surveillance for influenza and other respiratory pathogens. Our previous research has shown that influenza activity resurged in Russia during the 2021–2022 period, although it was of moderate intensity [9]. Based on the situation in the Southern Hemisphere and globally in early autumn 2022, we anticipated an influenza A(H3N2) season with circulation of influenza B towards the end. However, the situation in the Russian Federation differed significantly from the majority of countries in the WHO European region. While many countries experienced an A(H3N2) influenza season [10], the Russian Federation faced a completely distinct scenario. In this study, we provide a concise overview of the intense past influenza epidemic and present data on population immunity and influenza vaccine effectiveness.

## 2. Materials and Methods

### 2.1. Epidemiological Surveillance

During the specified time frame, we revised the baseline for the incidence and hospitalization rates associated with influenza, taking into consideration data from the preceding six seasons, excluding the 2020–2021 season, when no influenza outbreaks were reported in Russia. We determined the pre-epidemic and post-epidemic baselines using the moving epidemics method [11]. Integrated non-sentinel and sentinel surveillance for influenza and SARS-CoV-2 was introduced in Russia in April–May 2020. Sentinel surveillance was conducted in strict accordance with the WHO guidance [12]. In order to show and compare influenza intensity in season 2022–2023 we also present epidemiological data from non-sentinel surveillance for eight consecutive seasons.

Influenza virus isolation, identification, and antigenic analysis were performed according to the WHO Manual [13] and appropriate guidelines approved in Russia [14]. Madin-Darby canine kidney cells (London line, kindly provided by the WHO CC World Influenza Centre, London, UK) were used for isolation of influenza A(H1N1)pdm09 and B viruses from rRT-PCR positive samples. Influenza A(H3N2) viruses were isolated in MDCK-Siat1 cells as described in [14]. Briefly, cells were grown in Nunc cell culture tubes (Nunc, Thermo Fisher Scientific, Waltham, MA, USA) to form a 90–95% confluent monolayer 1 day before inoculation. Cells were washed twice with minimal essential medium (MEM) containing 2 μg/mL of N-tosyl-L-phenylalanine chloromethyl ketone-treated (TPCK)-trypsin and penicillin-streptomycin (100 units and 100 µg/mL, respectively), and 200 μL of virus-containing media was inoculated into each tube. Tubes were kept at 36 °C for 40 min to allow virus absorption. Then, 1.8 mL of virus growth media was added [MEM containing 2 μg of TPCK-trypsin and penicillin-streptomycin, bovine albumin fraction V (2.6 mL per 100 mL, Sigma Aldrich, Saint Louis, MO, USA), HEPES buffer (1.6 mL, Sigma-Aldrich)]. Tubes were kept for 3–6 days at 36 °C and monitored daily for progression of CPE. Once CPE was detected, the tubes were frozen at −80 °C, thawed, and after centrifugation the hemagglutination assay with human (group O) red blood cell suspension (0.75%) was performed to determine viral titers. Influenza viruses isolated in MDCK or MDCK-SIAT1 [15] during the epidemic period 2022–2023 were characterized antigenically in hemagglutination inhibition assay (HI) or microneutralization assay (MN) with the strain-specific post-infectious ferret antisera or strain-specific hyperimmune rat antisera to the reference influenza viruses [16].

rRT-PCR (PCR) monitoring for the etiology of SARI, ILI, and ARI cases by weekly analysis of data obtained in routine and sentinel influenza surveillance was performed. RNA was isolated from the clinical samples using the AmpliSense^®^RIBO-prep kit (InterLabService, Moscow, Russia) or the RNeasy Mini kit (QIAGEN, Hilden, Germany). Reverse transcription of RNA was performed using the Reverta-L kit (InterLabService, Russia) or the OneStep RT-PCR Kit (QIAGEN) with CDC primers and probes. PCR for influenza A and B viruses was performed using the AmpliSense^®^ Influenza virus A/B-FL kit (InterLabService, Russia), subtype identification was performed using AmpliSense^®^ Influenza virus A-FL kit subtyping H1N1, H3N2 (InterLabService, Russia). Multiplex AmpliSense^®^ kit for detection of influenza and SARS CoV-2 was used. Detection of other respiratory viruses was performed using the multiplex AmpliSense^®^ ARVI-screen-FL kit (InterLabService, Russia). All real-time PCRs were carried out using a Rotor-Gene 6000 (Corbett Research, Mortlake, Australia) or a CFX96 Touch™ Real-Time PCR Detection System (BIO-RAD, Hercules, CA, USA).

### 2.2. Population Immunity Investigation

To assess the level of population immunity to influenza viruses, we assessed humoral immunity twice a year—in the pre-epidemic period (October–November) and post-epidemic period (April–May) of 2019–2022. The sera from adult blood donors aged 18–60 years with the accompanying information on their age and vaccination status were obtained from the Regional Base Laboratories (usually not less than 100 sera from each RBL). HI assay was carried out in accordance with WHO guidelines using 0.75% chicken erythrocytes [13]. The donor sera were preliminary diluted 10 times in PBS, heated at 56 °C, and treated with RDE in accordance with the Manufacturer’s instructions (Denka Seiken) and approved Guidelines [17]. In line with WHO recommendations for influenza vaccine composition, we used specific strains in the HI assay for each influenza virus subtype. For influenza A(H1N1)pdm09, we utilized the strains A/Brisbane/02/2018 in the 2019–2020 season, A/Guangdong-Maonan/SWL1536/2019 in the 2020–2021 season, and A/Victoria/2570/2019 in the 2021–2022 season. To detect antibodies against influenza A(H3N2), we used the strains A/Kansas/14/2017 in the 2019–2020 season, A Hong Kong/2671/2019 in the 2020–2021 season, A/Cambodia/e0826360/20-like virus in the 2021–2022 season, and A/Darwin/9/2021 in the autumn of 2022. For the study of immunity to influenza B/Victoria virus, we used the strain B/Colorado/06/2017 in the 2019–2020 season and B/Washington/02/2019 in the subsequent two seasons. A protective antibody titer of 1:40 or higher was considered when determining the percentage of seropositive donors for influenza.

### 2.3. NGS Sequencing

Libraries for Illumina sequencing were prepared using Illumina DNA Prep library preparation kit (Illumina, San Diego, CA, USA) and then sequenced on NextSeq2000 instrument (Illumina, USA) with NextSeq Reagent kit P2 100-cycle. FastQC software was used for sequence data quality assessment. Trimmomatic was used for quality data trimming. Reads were mapped onto reference sequence using BWA. Consensus sequence was obtained using Samtools mpileup and Ivar software. Libraries for Oxford Nanopore sequencing were prepared using DNA Ligation Sequence kit SQK-LSK109 with Native Barcoding Expansion EXP-NBD196 (Oxford Nanopore, Oxford, United Kingdom). Sequencing was performed on MinIon instrument (Oxford Nanopore, United Kingdom) with R9.4.1 flowcell. Guppy software was used for basecalling and data quality trimming. Reads were mapped onto reference sequence using Minimap2. Consensus sequence was obtained using Medaka software.

### 2.4. Phylogenetic Analysis

Phylogenetic trees were built for the HA and NA genome segments. Human influenza virus sequences were downloaded from the GISAID for the analysis. The following criteria were used: (1) full genome segments; (2) virus type A or B; (3) A(H1N1), A(H1N1)pdm09, or A(H3N2) subtype; and (4) reference of vaccine strain. Sequences for each included strain were aligned using the Muscle application within MEGA7.0 software. Maximum-likelihood phylogenetic trees were built using MEGA7.0 with 1000 bootstraps using the general time-reversible model (GTR) with gamma rate categories.

### 2.5. Vaccine Effectiveness (VE)

VE calculations were performed according to the WHO manual [18] using the test-negative design approach.

### 2.6. Statistical Analysis

Statistical analyses were performed using Statistica 10.0 nonparametric criteria, bivariate method chi-square (*p*-values less than 0.05 were considered to be statistically significant). All confidence intervals were 95%.

### 2.7. Ethical Aspects of the Study

The investigation was conducted in compliance with the principles of Good Clinical Practice (GCP) and in accordance with the regulations set by the Federal Service for Surveillance in the Field of Consumer Rights Protection and Human Well-Being (Rospotrebnadzor). The study received approval from the Local Ethics Committees prior to its commencement. Patient consent to participate in the study was obtained as a mandatory requirement in all hospitals, polyclinics, and donor centers.

## 3. Results

### 3.1. PCR Monitoring of Influenza Viruses Spread

Weekly monitoring of influenza virus circulation was carried out by performing PCR detection of the virus in clinical samples from 220,067 patients with ILI in non-sentinel surveillance. The results of this monitoring revealed an unusually early onset and rapid growth of influenza activity at the beginning of the 2022–2023 season. The first cases of influenza A(H1N1)pdm09 and A(H3N2) viruses were detected in week 41.2022, followed by a rapid increase in the number of influenza A(H1N1)pdm09 cases starting from week 45.2022. Influenza B virus activity slowly began to increase from week 45.2022, and it surpassed influenza A cases from week 4.2023 onwards, peaking in week 7.2023. The total detection rate of influenza viruses exceeded the 10% epidemic threshold as early as week 48 of 2022. The peak of PCR-positive cases for influenza was observed in week 51.2022, with 3399 cases (28.2%) per week reported by the two NICs and collaborating Regional Base Laboratories (RBL). Throughout the season, influenza A(H1N1)pdm09 virus dominated, accounting for 56.4% of all cases, while influenza B viruses averaged 34.3%. Influenza A(H3N2) virus activity remained low, estimated at 0.6% of positive cases. In 8.75% of cases, influenza A virus was not subtyped (Figure 1).

A similar trend was observed in the sentinel surveillance system, which involved 18 hospitals located in different geographically distant cities in Russia. Overall, in the season, 1576 patients with Severe Acute Respiratory Infection (SARI) were admitted. In the first half of the epidemic, the dominant virus was influenza A(H1N1)pdm09, while in the second half, the influenza B virus became more prevalent. Cases of influenza A(H3N2) among SARI patients were rare and mostly occurred towards the end of the season (Figure 2).

According to PCR data obtained, the duration of the influenza epidemic in Russia last season was 13 weeks. The average percentage of influenza virus detection for the entire period was estimated to be 11.5%. However, during the peak of Influenza-Like Illness (ILI) incidence in routine surveillance, this percentage was 2.5 times higher, ranging from 28.2% to 29.6%. The percentage of influenza virus detection was especially high (52.8%) during the peak of the epidemic in the sentinel surveillance system.

### 3.2. The Geographic Spread of Influenza Viruses by Type/Subtype

The contribution of A(H1N1)pdm09, A(H3N2), and B (Victoria lineage) viruses in morbidity was evaluated to understand pathways and geographical spread of influenza viruses in the European part of the country, Siberia and Far East over time. This is presented below in a series of maps. The first cases of influenza A(H1N1)pdm09 and A(H3N2) viruses were detected on week 41.2022 in two megacities (Moscow and St. Petersburg, respectively, that together comprise 12% of the population). From there, the viruses spread all over the European part of the country, and from the week 48.2022 to the cities of the Urals, Siberia, and the Far East. The first influenza B cases were detected in the Ural region (Yekaterinburg) on week 44.2022. Then, the viruses spread to the European part of the country, the Far East, and Siberia. Co-circulation of influenza A(H1N1)pdm09 and B was reported widely. At the beginning of 2023, further expansion of the geography of influenza B virus circulation became evident and from week 10 to the end of the season it began to prevail over influenza A(H1N1)pdm09 viruses (Appendix A).

The comparative analysis of influenza activity over the past eight seasons has revealed an unprecedentedly rapid increase in both ILI incidence and the number of PCR-confirmed influenza cases during the peak of the 2022–2023 epidemic season. The intensity of the epidemic surpassed the previously recorded “very high level” index, which had never been observed in the previous seven years. Additionally, the duration of the influenza epidemic in Russia was notably prolonged compared to previous seasons (Figure 3).

It is worth noting that the most intense and rapidly developing epidemics in Russia were associated with the dominance of the influenza A(H1N1)pdm09 virus during the 2015–2016 and 2022–2023 seasons. On the other hand, epidemics caused by the influenza A(H3N2) virus during the 2016–2017 and 2021–2022 seasons, as well as mixed-type epidemics (2017–2018, 2018–2019, and 2019–2020), developed gradually and were moderate in intensity.

The contribution of the influenza B virus to ILI incidence was less pronounced in all seasons and typically recorded towards the second half of the epidemics. However, in the last epidemic, there was a long-term excess of BL-ILI incidence, which could be partly attributed to the circulation of influenza B viruses (Appendix A).

### 3.3. Antigenic Analysis of Isolated Influenza Viruses

The characterization of 777 influenza A(H1N1)pdm09 strains obtained from different cities of the country, performed by two NICs has shown that all of them were A/Victoria/2570/2019-like and were recognized well (within twofold of the homologous titres) by antisera raised against this vaccine strain. The tested viruses were also recognized well by antisera raised against A/Denmark/3280/2019 and A/Wisconsin/588/2019 reference strains, which belonged to the same genetic group 6B.1A.5a2 as the vaccine strain. Most of the test viruses were recognized less well by the antisera raised against the previous reference strain A/Guangdong-Maonan/SWL1536/2019 (between 1/4- and 1/16-fold of homologous titre); meanwhile, they were recognized with fourfold of the homologous titre by the antiserum raised against to A/Hong Kong/110/2019 of the group 6B.1A.2, which had the substitution N156K, specific for the more recent groups; however, all the strains were poorly recognized by the antiserum to an earlier strain A/Brisbane/02/2018. The antigenic map of the A(H1N1)pdm09 viruses illustrated these data: all viruses of the current season form a compact group near vaccine strain A/Victoria/2570/2019, being just in the center of this group. At the same time, the previous reference strains A/Guandong-Maonan/SWL1536/2019 and especially A/Brisbane/02/2018 are positioned distantly of this group as well as Russian isolates of the 2018–2019 and 2019–2020 season (Figure 4).

### 3.4. Antigenic Analysis of Influenza A(H3N2) Viruses

The viruses of this sub-type were isolated only in three cities of the country and were in the minority over the whole season. All 27 tested viruses were recognized well (within 1/4-fold of the homologous virus titer) by ferret and rat antisera raised against the egg-grown reference strains A/Darwin/06/2021 and A/Darwin/09/2021, the recommended vaccine strains for the season 2022–2023. They were recognized less well by the antisera raised against the previous vaccine and reference strains A/Cambodia/925256/2020 and A/Cambodia/e826360/2020 (up to 8–16-fold of homologous titre, see Appendix A).

### 3.5. Influenza B Viruses

Most of the 243 influenza B viruses (all of Victoria lineage) tested were recognized well (within 2-fold of homologous titre) by the antiserum raised against the B/Austria/1359417/2021. The antigenic map depicts dense clustering of the strains of 2022–2023 season isolation as well as of the previous season around the vaccine strain B/Austria/1359417/2021 (both egg- and MDCK variants) within 1–2 squares observed while being distanced from the strains of previous seasons, such as B/Colorado/06/2017, B/Washington/02/2019, B/Croatia/779/2019 and B/Cote d’Ivoire/948/2020 (Figure 5).

Evaluation of susceptibility of influenza isolates to antivirals has shown that 603 of 606 isolates, obtained from 22 cities of Russia (including 500 strains of influenza A(H1N1pdm09), 25 influenza A (H3N2) viruses and 81 influenza B viruses), tested for susceptibility to neuraminidase inhibitors by MUNANA fluorescent test, were susceptible to oseltamivir and zanamivir (registered IC_50_ was in the range 0.3–6.1 µM). Two influenza viruses, namely A/Novosibirsk/7.271/2022 (H1N1)pdm09 and B/Novosibirsk/7.308/2022, had reduced susceptibility to oseltamivir; one influenza virus, the strain A/St. Petersburg/NIIG-125/2023 (H1N1)pdm09, was resistant to both neuraminidase inhibitors.

Full genome sequencing of influenza viruses circulating in Russia was conducted by the NIC at the Smorodintsev Research Institute of Influenza, resulting in the deposition of 2237 sequences in GISAID.

Of the 1723 influenza A(H1N1)pdm09 viruses sequenced, all belonged to clade 6B.1A.5a.2 (A/Victoria/2570/2019 vaccine virus-like), with characteristic amino acid substitutions in HA, including K130N, N156K, L161I, V250A, and E506D. Additionally, all sequenced viruses encoded K54Q, A186T, Q189E, E224A, R259K, and K308R in HA1. Among this subgroup, the HA of three viruses from Saint Petersburg were genetically closely related to A/Sydney/05/2021 (vaccine strain for the Southern Hemisphere for 2023), exhibiting D94N in HA with additional substitutions in HA1 (T120A, T277A, T278S). The remaining viruses were similar to A/Victoria/2897/2022 (the WHO recommended vaccine virus for the 2023–2024 northern hemisphere influenza season).

Most of the viruses possessed one amino acid substitution in antigenic site Sb compared to the vaccine strain A/Victoria/2570/2019 (Figure 6). Nine viruses from six regions of the Russian Federation had the H275Y substitution in the NA gene, a well-known genetic marker of oseltamivir resistance. Approximately half of the global dataset of influenza viruses from the 2022–2023 season in GISAID (1295 out of 3542) had two additional substitutions in antigenic site Ca2 (P137S, K142R), but the prevalence of such viruses in Russia appeared to be relatively low (<1%). The majority of influenza A(H1N1)pdm09 viruses from Russia formed a group characterized by several homoplastic synonymous nucleotide mutations (753 G>A [824], 1602 C>T [1673]), among others.

### 3.6. Genetic Analysis of Influenza A(H3N2) Viruses

A total of 81 influenza A(H3N2) full genomes were obtained and phylogenetically could be split into four subclades. The largest and genetically homogenous group belonged to 3C.2a1b.2a.2b subclade and differed from the vaccine virus A/Darwin/9/2021 by substitutions in HA: E50K, F79V, I140K, T135A, and S262N. Furthermore, all Russian viruses of this group had R269K in HA1 (Appendix A). The next group consisted of 13 viruses collected in December 2022, February, and March 2023 was A/Darwin/9/2021-like (subclade 3C.2a1b.2a.2a.1b) with substitutions in HA: I140K и R299K. The third group of 3C.2a1b.2a.2a.3a.1 subclade included two viruses collected in St. Petersburg in November 2022 with substitutions in HA: E50K, I140K, and I223V. The last group of viruses collected in Ural Federal District in February 2023 belonged to 3C.2a1b.2a.2a.3b subclade (A/Sydney/732/2022-like) with substitutions I140M and G62E in HA.

None of the influenza A(H3N2) viruses sequenced had markers of resistance to NA inhibitors in NA.

### 3.7. Genetic Analysis of Influenza B Viruses

A total of 433 influenza B viruses (Victorian lineage) were sequenced during the season. All Russian isolates belonged to the subclade V1A.3a.2 (B/Austria/1359417/2021-like) encoding the HA1 substitutions A127T, P144L and K203R. This subclade could be further divided into two subgroups. A large subgroup contains viruses with HA genes encoding the HA1 substitution D197E (B/Stockholm/5/2021-like), another subgroup comprises viruses with E128K, A154E, and S208P in HA1 (Figure 7). None of the sequenced viruses had markers of resistance to NA inhibitors in NA.

### 3.8. The Effectiveness of Influenza Vaccines

Assessment of the effectiveness of influenza vaccines (IVE) in the context of the current data on full matching of circulating influenza viruses and the strains included in the composition of influenza vaccines, as shown by us at the antigenic and genetic level, was of special interest. To perform this evaluation, we used the data from sentinel surveillance where the patients have laboratory-confirmed diagnoses. We used the test-negative case-control approach as described in [17]. This season, the overall vaccination coverage among PCR-confirmed hospitalized patients with severe acute respiratory infection (SARI) and among influenza-confirmed influenza-like illness/acute respiratory illness (ILI/ARI) cases was low (0.7% and 6.6%, respectively). The IVE estimated in the sentinel surveillance system was evaluated as 92.7% in the prevention of SARI patient admissions and 54.7% in the prevention of influenza among outpatients with ILI/ARI (average 80%, Table 1).

### 3.9. Determination of the Population Immunity Level to Influenza Viruses

We performed an investigation of population immunity to understand the possible reasons for the development of such an intense influenza epidemic in the last season and find whether it was related to antigenic drift of influenza A(H1N1)pdm09 and B viruses alone or also to a decrease in the level of population immunity. To clarify this issue, we compared the level of humoral population immunity (PI) among adults in the four pre-epidemic (autumnal) and three post-epidemic (vernal) periods. The HI data has shown a significant decrease in population immunity to influenza viruses over the past three years, after the spread of the SARS-CoV-2 pandemic. The geometric mean titers (GMT) of antibodies to influenza A(H1N1)pdm09, A(H3N2), and B/Victoria viruses during the autumnal, pre-pandemic period decreased more than three times: from 38.3, 38.2, and 24.4 in 2019 to 10.1, 12.8, and 2.8, respectively, in 2022 (Figure 8).

Furthermore, we observed a significant decrease in the percentage of individuals with a protective antibody titer (≥1:40) from the pre-epidemic period of the 2019–2020 season to the pre-epidemic period of the 2022–2023 season for influenza A(H1N1)pdm09 (from 59.6% to 34.5%), influenza A(H3N2) (from 57.2% to 32.5%), and influenza B/Victoria lineage (from 47.3% to 10.6%). Such decline in population immunity created favorable conditions for such widespread intense circulation of the influenza A(H1N1)pdm09 and influenza B/Victoria viruses during the 2022–2023 season in Russia.

## 4. Discussion

The SARS-CoV-2 pandemic had a dramatic impact on the circulation of influenza and other respiratory viruses at the local and global levels [19], which, after a one-year break, returned rapidly to global circulation. Epidemics of moderate intensity with the predominant participation of the A(H3N2) virus were registered in the previous season (2021–2022) in most of the countries with a low impact of influenza B virus [20]. In the 2022–2023 season, an unexpectedly high-intensity epidemic began to develop rapidly in Russia with influenza A(H1N1)pdm09 dominating over other types and subtypes. This distinguished the epidemic in Russia from most other countries of the Northern Hemisphere; whereas, in the previous season, the influenza A(H3N2) virus remained the dominant circulating virus, and influenza A(H1N1)pdm09 virus started to be registered only by the end of the epidemic [10]. This indicates the specificity of the etiology of influenza in different countries and determines the need for global surveillance of this pathogen in order to understand the evolutionary directions and ways of the spread of influenza viruses.

Influenza A(H1N1)pdm09 viruses circulating in Russia underwent a significant antigenic drift from previous vaccine strains A/Guangdong-Maonan/SWL1536/2019 and A/Brisbane/02/2018 matching closely the A/Victoria/2570/2019 strain introduced into the vaccine composition for the Northern Hemisphere season 2022–2023. All tested strains belonged to the same genetic group 6B.1A.5a2, which was also lately isolated in Europe, the USA, and other locations [21]. This antigenic uniformity of the A(H1N1)pdm09 strains in the country had a positive effect on the effectiveness of influenza vaccines in the prevention of SARI and ILI/ARI, which was higher than in previous seasons due to the (partial) strain mismatch. It is well known that influenza vaccine effectiveness is highly dependent on the matching of strains in influenza vaccines and actually circulating viruses [22]. In recent years, many tools have been proposed to contribute to the vaccine selection process including short-term modeling strategies and visualizations [4,23,24] but none of them currently can predict the vaccine strains without ongoing global surveillance activities and massive antigenic and genetic analysis work conducted by the WHO CCs and the NICs [25].

An objective analysis of the influenza virus strains selected for vaccine composition leads to the conclusion that the annual investigation of the antigenic structure of viruses is essentially retrospective: it is completed in February while the vaccination starts at least 6 months later. This time gap provides an essential period for the virus to accumulate point mutations that may provide an effective immune escape.

The number of such escape mutants that occur at the population level can be quite large, but the question of which one will become widespread in the next season has not yet been answered.

Meanwhile, apart from studying the evolution of the virus directly the other way to understand the possible impact of the new variant is to assess the population immunity. Each influenza infection and vaccination (especially with the live attenuated vaccines) provide complex immunity with at least some cross-protection against variant viruses. Our study has shown that the population immunity of the adult population was diminished before the start of the last epidemic with a more profound decrease for influenza A(H1N1)pdm09 and influenza B. Recently, it was found that a 10–60% increase in population susceptibility might lead to a maximum of onefold to fivefold rise in peak magnitude and a onefold to fourfold rise in the epidemic size for the upcoming influenza season [26].

In the past two decades, multidirectional work has been carried out to create universal influenza vaccines based on conservative viral determinants [27,28,29] that theoretically should drastically improve the population immunity against many if not all influenza A viruses; however, none of them meets the requirements of high protection either against contemporary epidemic viruses or potentially pandemic influenza A virus, leaving the need for an annual update of current influenza vaccines.

To address the issue of vaccine strain mismatch, it is crucial to enhance global surveillance efforts for influenza viruses. This includes continuous monitoring of viral antigenic characteristics, genetic sequencing, and analysis of viral evolution patterns. By identifying and tracking emerging strains, public health officials can make more informed decisions regarding vaccine composition for future seasons.

The WHO GISRS and its partners are continually working to identify improvements and harness new technologies to strengthen and sustain collaboration. WHO will continue to play the central role of coordinating worldwide expertise to meet the increasing public health need for influenza vaccines and support efforts to improve the vaccine virus selection methodology, including through the convening of periodic international consultations [30]. The recent epidemic in Russia highlights the importance of global surveillance to understand the specific etiology and spread of influenza viruses in different countries.

### Limitations

The analysis presented in the paper contains certain limitations, the most important of which are the limited geographic representation of sentinel surveillance data. Only 10 major cities located in all federal districts were providing sentinel surveillance data. Limited by the incomplete geographical representativeness, we may have missed some circulation of influenza A(H1N1)pdm09, A(H3N2), or influenza B. The overall number of patients included in the sentinel surveillance data was limited, and their distribution across different age groups was uneven. Consequently, these factors could have influenced the calculation accuracy of the vaccine effectiveness (VE), as well as hindered the identification of the specific pathogen in each patient within the study. As a result, it is possible that we may have overlooked certain cases of influenza positivity, as well as other respiratory infections. Assessment of population immunity was only available for healthy adult blood donors which of cause is not representative of the general population, especially not for the highly affected young children and the elderly population.

## Figures and Tables

**Figure 1 viruses-15-01780-f001:**
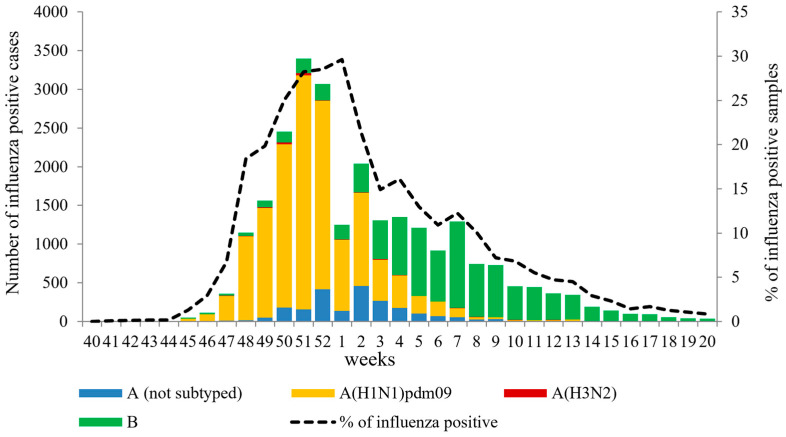
Weekly PCR detection of influenza viruses in non-sentinel surveillance system in Russia for the period from week 40.2022 to week 20.2023.

**Figure 2 viruses-15-01780-f002:**
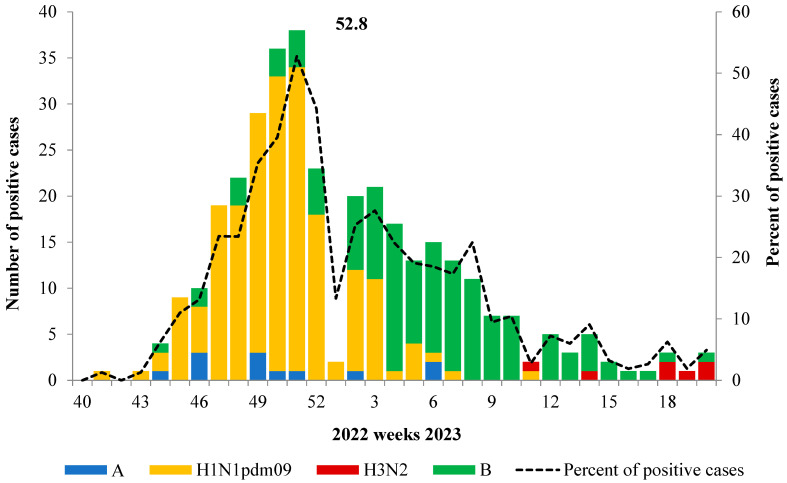
Influenza virus detection among SARI patients admitted to hospitals in sentinel surveillance system.

**Figure 3 viruses-15-01780-f003:**
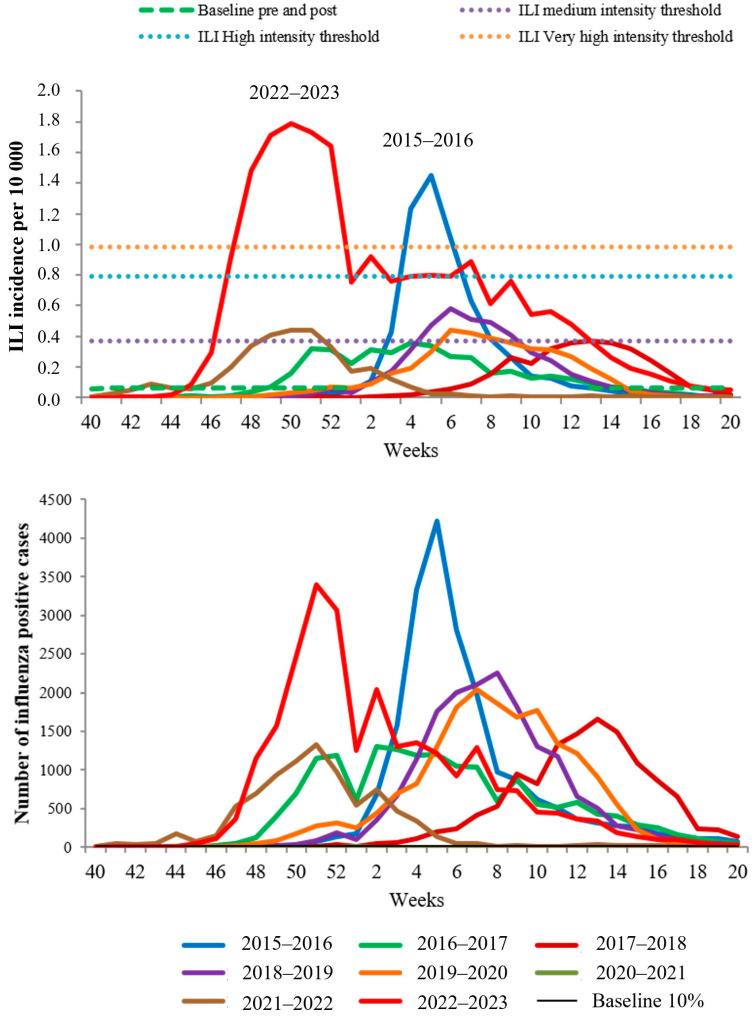
Comparative data of influenza incidence and PCR monitoring in non-sentinel surveillance for seven consecutive seasons.

**Figure 4 viruses-15-01780-f004:**
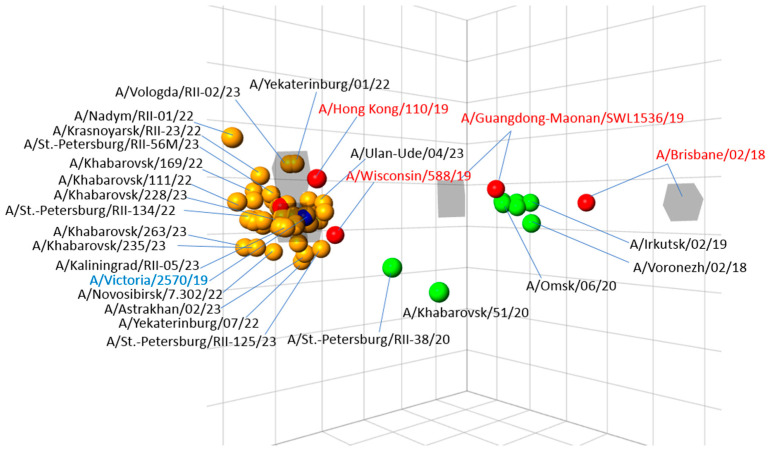
Three-dimensional antigenic map of influenza A(H1N1)pdm09 viruses isolated in Russia in the epidemic season 2022–2023. Note: Red circles—reference antigens; orange circles—Russian isolates of the current season; green circles—Russian isolates of the previous seasons; grey squares—ferret post-infectious antisera.

**Figure 5 viruses-15-01780-f005:**
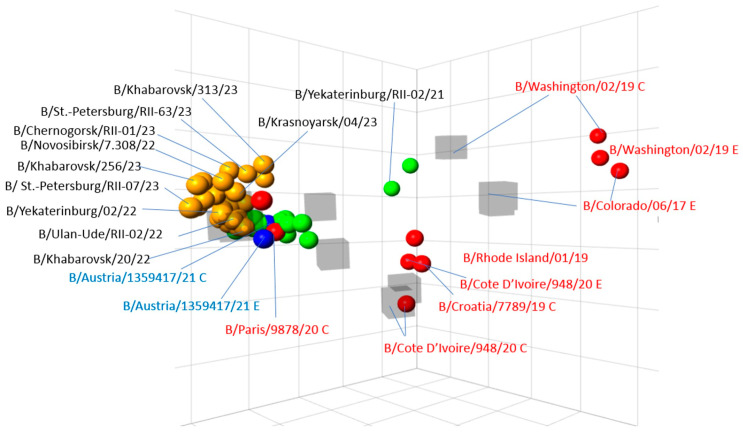
Three-dimensional antigenic map of influenza B viruses (Victoria lineage) isolated in Russia in the epidemic season 2022–2023. Note: designations as in Figure 6.

**Figure 6 viruses-15-01780-f006:**
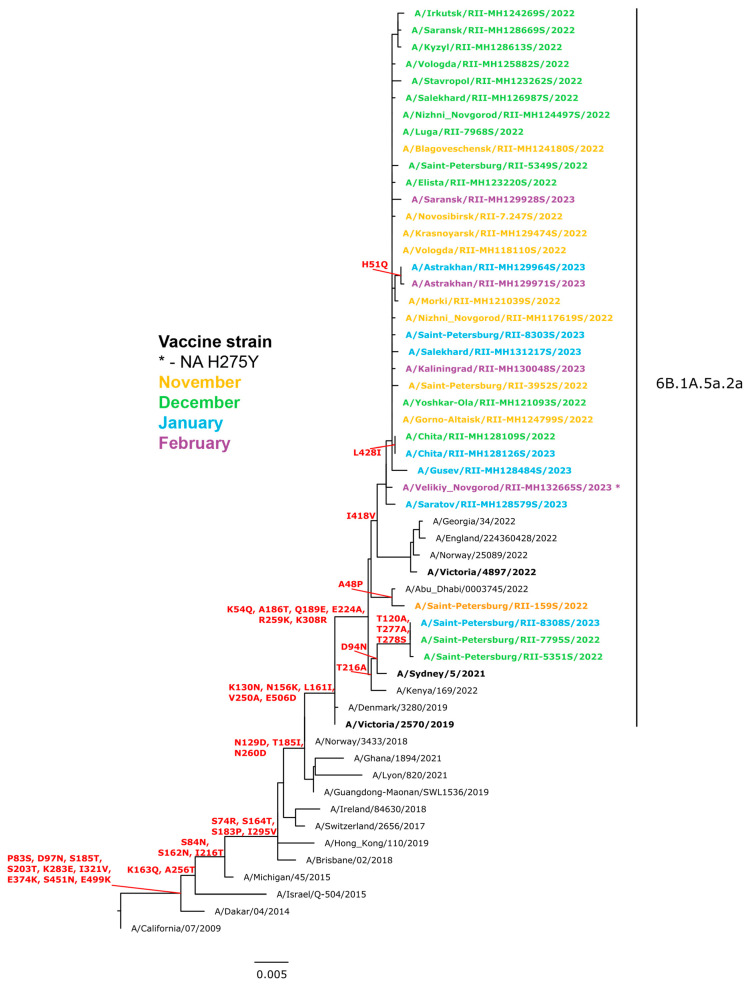
Phylogenetic comparison of HA genes of influenza A(H1N1)pdm09 viruses (ML tree, GTR+GAMMA, constructed with RaxML). Note: Viruses bearing the H275Y mutation in NA (known marker of oseltamivir resistance) are marked with an asterisk (*) at the end of the name. Not all resistant strains are shown in the tree.

**Figure 7 viruses-15-01780-f007:**
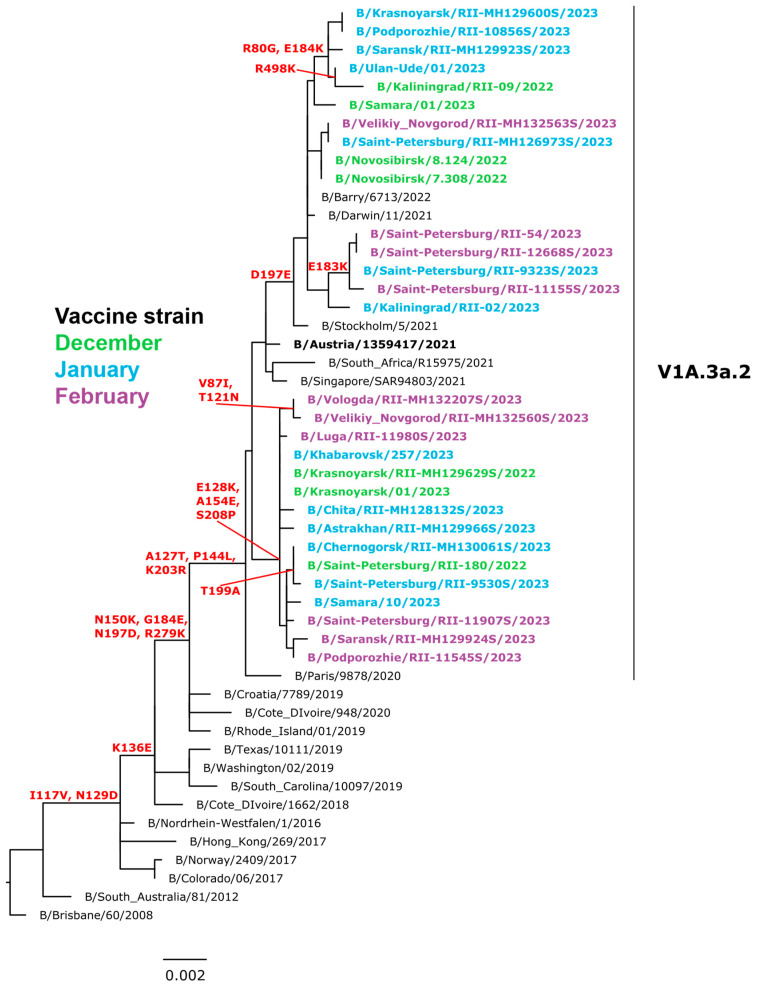
Phylogenetic comparison of HA genes of influenza B (Victoria lineage) viruses (ML tree, GTR+GAMMA, constructed with RaxML).

**Figure 8 viruses-15-01780-f008:**
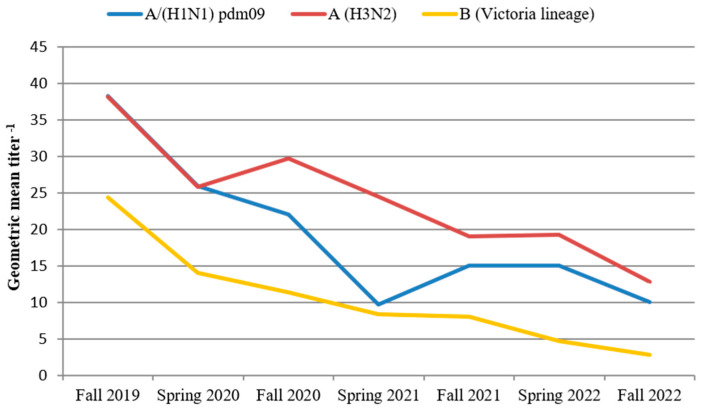
Changes in the level of population humoral immunity to influenza viruses in adults for the period from 2019 to 2022.

**Table 1 viruses-15-01780-t001:** Influenza vaccine effectiveness determined for the period from week 40.2022 to week 10. 2023 in prevention of influenza among patients with SARI and outpatients with ILI/ARI.

Category	Vaccinated		Not Vaccinated		Risk among Vaccinated	Risk among Not Vaccinated	Odds Ratio (OR) ad/bc	VE 1-OR	CI
	Flu (+)	Flu (−)	Flu (+)	Flu (−)					
	(a)	(b)	(c)	(d)					
SARI	2	107	312	1210	1.83	20.50	0.07	92.75	(70; 98)
ILI/ARI	8	144	112	914	5.26	10.92	0.45	54.66	(5.12; 78.4)
Average	10	251	424	2124	3.8	16.6	0.2	80.04	(62.13; 89.48)

## Data Availability

All data on influenza epidemiological and virological surveillance in Russia in the season 2022–2023 and all the previous seasons described in the article are freely available as the National weekly influenza bulletins at the website of Smorodintsev Research Institute of Influenza (https://www.influenza.spb.ru/en/influenza_surveillance_system_in_russia/epidemic_situation/, accessed on 29 June 2023).

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
