# Peer review of "Assessing the Intense Influenza A(H1N1)pdm09 Epidemic and Vaccine Effectiveness in the Post-COVID Season in the Russian Federation"

_viruses, 2023, doi:10.3390/v15081780_

Round 1

Reviewer 1 Report

This is an interesting study of the influenza season in part of Russia during 2022-2023 influenza season after the SARS-COV-2 pandemic.   It goes through many different aspects of the viruses circulating, immunity detected in this particular population, antiviral sensitivity, and discusses vaccine effectiveness.  All of this was fairly well done, but the vaccine effectiveness calculations do not seem quite right or have significant limitations that are not discussed.  

The authors report a >92% vaccine effectiveness abut I am not sure they did this correctly.   Although their number is true using the method of calculation they used, this method is very limited and does not take into account numerous factors that could be playing a role.  

1. The numbers are quite small, and it should be pointed out that this is a significant limitation in evaluating effectiveness.

2.  After a large scale pandemic with a coronavirus that primarily negative effected the elderly who are also the most susceptible to severe disease, and had significant mortality during the COVID pandemic, evaluating vaccine effectiveness at preventing severe disease to influenza at the same time could lead to significant confounding of the data as even our molecular testing is imperfect in detecting these respiratory viruses.

3.  They way they calculated the effectiveness was to compare people with SARI or ILI who were FLU+ to  people who were negative for Flu, but those people really have nothing to do with the vaccine effectiveness at preventing severe disease in this study.  They need to compare people who had flu with severe disease to those with regular ILI.  If you do the calculation that way, using odds rations that compare SARI to ILI in the vexed and unvaxxed group, you get a VE closer to 73% at preventing severe disease in flu positive patients, and 20% VE in preventing SARI in Flu negative patients which is more in line with what one would expect from historical data even with a perfectly matching vaccine.   I think this is a more accurate way to calculate this VE in this instance and the 92% number is way off.     They way they did it COVID and other severe respiratory disease is confounding their numbers.

The manuscript should be edited either to remove the VE discussion or it should be recalculated in a more accurate method and limitations discussed of this assessment.  If that is done, then I think this is a good manuscript.

Author Response

Dear Reviewer 1,

Thank you very much for detailed reviewing our manuscript. Please find below the point-by-point answers to your comments and proposals.

  1. We fully agree that the number of the patients under sentinel surveillance was rather small, and is a significant limitation in evaluating vaccine effectiveness.
  2. Of course, some of COVID-19 and influenza cases could not be detected due to low viral load in the samples or because of the late stages of infection. However, it should be noted that all patients were tested for influenza, SARS-CoV-2 and other respiratory viruses in PCR which is a powerful method and has a high sensitivity.
  3. In terms of the method to calculate vaccine effectiveness we followed the widely used design described in the WHO manual. One of options for selection of controls recommended by WHO for annual VE estimation is using test-negative design (TND, a special instance of case-control evaluation). The essence of the method is to use patients who meet the specimen collection criteria from the study protocol, are tested for influenza, and are found to have negative test results. In this approach, the target population for enrolment consists of all persons who seek care for a defined set of symptoms (ILI or SARI); cases are those with positive tests for influenza, and non-cases are those with negative test results. It is powerful for several reasons. Firstly, all cases and controls have sought care at the same facilities. Hence, cases and non-cases will generally have come from the same communities, reducing bias due to community-level variations in vaccine coverage. Secondly, cases and non-cases have all sought care for similar sets of symptoms. This reduces confounding due to differences in health-care seeking behavior between cases and non-cases, which is a major challenge to influenza VE studies. Vaccine status is typically collected and recorded at the time of specimen collection, prior to knowing the influenza test result, reducing the likelihood of differential exposure misclassification. Even with sensitivity for influenza detection as low as 70%, in the context of near-perfect specificity such as provided by influenza laboratory-confirmed by PCR, outcome misclassification has been shown to have negligible impact on VE estimates derived by the test-negative design. Moreover, the TND design has been approved and widely used by the I-MOVE consortium that published a lot of studies on VE in the WHO European region.

Despite these limitations, the study's calculation of VE for SARI patients appears to be conducted correctly, although the numbers obtained are not typical for influenza VE. We also added the confidence intervals to VE calculations that show the possible VE variation.

Kindly note, that we found a high level of antigenic similarity between the vaccine A(H1N1)pdm09 strain and the circulating strains in Russia. This high similarity should have contributed to a high VE, especially considering the dominance of A(H1N1)pdm09 in circulation.

The high intensity of the influenza epidemic can also be attributed to low vaccine coverage and a decrease in herd immunity. These factors may have further contributed to the spread of the virus.

Reviewer 2 Report

This study by dr. Sominina and co-workers “Assessing the Intense Influenza A(H1N1)pdm09 Epidemic and Vaccine Effectiveness in the Post-COVID Season in the Russian Federation” analyses the flu epidemic of the 2022-2023 season. The outbreak had an early onset and peaked in late 2022. Weekly monitoring of influenza virus circulation was carried out by performing PCR in clinical samples from 220,067 symptomatic patients.  Influenza A(H1N1)pdm09 virus was responsible for 56.4% of positive cases whereas  influenza B and Influenza A(H3N2) viruses were detected in 34.3% 0.6% of positive cases, respectively.

It is well known that the COVID-19 pandemic has had a significant impact on the circulation of human seasonal influenza viruses worldwide and has adversely affected influenza surveillance. So, despite the continued circulation of Omicron descendant lineages, influenza activity has resumed in full force. That extremely high circulation of the flu virus may have been due to a significant decrease in herd immunity. However, the flu vaccine has been shown to be highly effective in preventing hospitalization for severe acute respiratory illnesses caused by Flu. 

This study provides a concise overview of the recent influenza epidemic (2022-2023 season) in the Russian Federation by providing data on population immunity and influenza vaccine efficacy.

Despite some limitations described, such as the limited geographic representation of sentinel surveillance data, results of this study highlights the role of surveillance to understand the specific etiology and spread of influenza viruses in the different countries of the world.

General comments:

The paper is well structured, well written, and understandable to a specialist readership. The title clearly indicates the focus of the article and the Abstract section efficiently summarizes the contents of the paper. In the “Introduction” the context of the subject area is properly addressed to justify the study and the objective of the manuscript is clearly indicated. “Materials and methods” are suitable and the statistical analyses are well defined and appropriate to the design.Figures and Table are well designed and all necessary for understanding of the text. Conclusion provides interpretation of the results in the context of other evidence.

Minor comments

-       Summary

Please specify that SARI, ARI and ALI mean severe acute respiratory illness, acute respiratory illness and influenza-like illness, respectively

-Lines 105-106

Please change:

“Cells were washed twice with MEM media containing 2 μg of TPCK-trypsin and penicillin-streptomycin (10,000 units and 10 mg/ml, respectively),”

with

Cells were washed twice with minimal essential medium (MEM) containing 2 μg/ml of N-tosyl-L-phenylalanine chloromethyl ketone-treated (TPCK)-trypsin and penicillin-streptomycin (100 units and 100 µg/ml, respectively),

-Line 120

Please change:

cthe linical 

with

the clinical

-Line 236

Please remove the italic font

-Line 328 Figure 6

What does *-NA H275Y mean? What are the strains with the mutation?

-       Line 368 Table 1

In “Category” 

change ILI with ILI/ARI

-Line 371

Please remove “and”

English is of good quality

Author Response

Dear Reviewer 2,

Thank you very much for careful reviewing our manuscript. We took into account all your comments when preparing the final version of the manuscript, which will undoubtedly improve its quality.

Reviewer 3 Report

In this manuscript, the author evaluated the Influenza epidemic and vaccine effectiveness in the Post-COVID Season using sentinel and non-sentinel surveillance systems in the Russian Federation. The main finding of this study is that the influenza A(H1N1)pdm09 virus dominated positive cases during the period of 2022-2023, compared to the author's previous surveillance conducted in 2021-2022 (PMID: 36146716). This surveillance study holds significant importance, and I have a few minor suggestions for improvement.

1. It would be beneficial if the author could ensure consistency between Figure 1 and Figure 2 in terms of color, labeling, and lines used.

2. In Figure 1 and Figure 2, where the author mentions Influenza A without specifying the subtype, it would be valuable to include information on which specific subtypes of influenza A were found to infect the human population based on NGS sequencing data. This additional detail is crucial for potential influenza virus surveillance in the future.

3. It is necessary to provide a reference for the statement "Co-infections of influenza A(H1N1)pdm09 and B were reported widely."

4. The figure resolution should be improved in the revised version to ensure better clarity and readability.

Author Response

Dear Reviewer 3,

Thank you very much for careful reviewing our manuscript. Please find below the point-by-point answers to your comments and proposals.

  1. As per your first comment, consistency between Figure 1 and Figure 2 in terms of color, labeling, and lines used has been reached. Thank you for noticing.
  2. According to obtained NGS sequencing data only human influenza A(H1N1)pdm09, A(H3N2) and B/Victoria lineage circulated in Russia that season and no another subtypes were recognized in analysis of influenza viruses in clinical specimens from the investigated patients with SARI/ILI.. A small number of cases where subtypes of influenza A virus could not determined, due to an insufficient amount of RNA in the sample at Ct more than 30.
  3. “Co-infections” was the wrong word. It has been corrected for “co-circulation”.
  4. All the figures were updated to provide better clarity and readability.

Round 2

Reviewer 1 Report

Just because the WHO and others recommend or use a certain method to calculate VE in some context, that does not mean that it is the best way to calculate VE in all circumstances.  For the outcomes discussed in this manuscript comparing FLU+ to FLU negative hospitalized patients makes no sense in this context.  If you are evaluating prevention of severe illness then you need to look at everyone at risk of severe illness from that disease.  People not exposed to FLU or suffering from severe disease from another illness are not at risk of severe illness from influenza and have nothing to do with VE against influenza, especially with a pandemic causing SARI from coronavirus.  This method makes no sense in this context and I still believe this should either be recalculated or removed from the manuscript.  

Reviewer 2 Report

The manuscript has been improved.

Reviewer 3 Report

The authors addressed all of my comments.